# Influenza Vaccination Coverage among Older Adults with Hypertension in Shenzhen, China: A Cross-Sectional Survey during the COVID-19 Pandemic

**DOI:** 10.3390/vaccines9101105

**Published:** 2021-09-29

**Authors:** Qiushuang Li, Minyi Zhang, Hongbiao Chen, Fei Wu, Juxian Xian, Liting Zheng, Minyi Liang, He Cao, Xiaofeng Zhou, Zihao Gu, Qihui Lin, Qing Chen

**Affiliations:** 1Guangdong Provincial Key Laboratory of Tropical Disease Research, Department of Epidemiology, School of Public Health, Southern Medical University, Guangzhou 510515, China; qiushuang@smu.edu.cn (Q.L.); myalison@smu.edu.cn (M.Z.); wudiedie1996@smu.edu.cn (F.W.); juxian123@smu.edu.cn (J.X.); 3160090116@smu.edu.cn (L.Z.); lmy1065516308@smu.edu.cn (M.L.); 2Department of Epidemiology and Infectious Disease Control, Longhua Key Discipline of Public Health for the Prevention and Control of Infectious Diseases, Longhua Centre for Disease Control and Prevention, Shenzhen 518109, China; gesila2021@163.com (H.C.); luojingwei001@163.com (H.C.); ct502528528@163.com (X.Z.); 13711762051@163.com (Z.G.)

**Keywords:** older adults, hypertension, influenza vaccination coverage, COVID-19

## Abstract

Background: Older individuals with hypertension are at a high risk of being infected with influenza. However, there have been few studies investigating the influenza vaccination status among older people with hypertension. The present work aimed to estimate the vaccination coverage and determine the predictors of seasonal influenza vaccinations among hypertensive patients aged over 60 years in Shenzhen, China. Method: The study used data from an online cross-sectional survey that was conducted in Shenzhen City, China, in October 2020. Frequencies and proportions of all the variables including sociodemographic characteristics and health-related information were described and tabulated based on the influenza vaccination status. Bivariate and multivariable logistic regression analyses were used to identify independent predictors associated with the influenza vaccination. Results: A total of 5216 older people with hypertension aged above 60 years were recruited. Overall, only 4.7% had received an influenza vaccine in the latest influenza season. Using the action toward being vaccinated as the primary outcome, the multivariable regression analysis showed that participants aged over 80 years (aOR 2.957, 95% CI: 1.784–4.900), obtaining higher education levels (aOR 1.424, 95% CI: 1.060–1.914 for high school, aOR 1.681, 95% CI: 1.066–2.650 for college or above), living with a partner (aOR 1.432, 95% CI: 1.068–1.920), using a family doctor (aOR 2.275, 95% CI: 1.744–2.968), and taking a physical examination 1–2 and ≥3 times each year (aOR 2.107, 95% CI: 1.601–2.772 and aOR 2.118, 95% CI: 1.083–4.143, respectively) were more likely to be vaccinated. In contrast, smokers had less likelihood of having the influenza vaccination than non-smokers (aOR 1.829, 95% CI: 1.208–2.767). Conclusions: The coverage rate of influenza vaccinations is far away from optimistic among older adults with hypertension. Additional works should be undertaken immediately to improve the influenza vaccination status.

## 1. Introduction

Influenza is an acute respiratory illness caused by the influenza virus. Older patients with hypertension are regarded as a high-risk group of being infected due to the higher possibility of a compromised immune system [1,2,3]. Significant morbidity and mortality have been reported worldwide because of acute complications developed from influenza such as encephalitis and pneumonia [3,4]. In the United States (US), 70.0% to 85.0% of influenza-related deaths occurred in people aged over 65 years and influenza-related hospitalizations among older people accounted for 50.0% to 70.0% [2]. In China, an average occurrence of 88,100 influenza-related deaths is reported each year, of which approximately 80.0% of deaths are related to the elderly [5].

As a vaccination is one of the most potent weapons against infectious diseases, the use of a seasonal influenza vaccine is recognized as the most beneficial measure to guard against an influenza infection. Previous studies have shown that influenza vaccinations significantly reduce influenza-related mortality and hospitalization among older patients with hypertension [6,7]. From the results of a meta-analysis, the vaccine efficacy for preventing hospital deaths due to influenza ranged from 31% to 65% [8]. Additionally, a study conducted in Denmark from 2007 to 2016 indicated that an influenza vaccination was associated with an 18% reduced risk of death during influenza seasons among patients with hypertension [9]. A lower vaccination coverage has been reported in developing countries than in developed settings [10,11,12,13,14]. Influenza vaccination coverage among elderly people with hypertension has been also surveyed in several regions. For instance, it ranged between 26% and 36% in Denmark during nine influenza seasons [9]. Kyeong and colleagues reported that 87.3% and 90.2% of male and female older adults with hypertension were vaccinated against influenza, respectively [12]. However, there are few investigations on this topic in China.

The Chinese Center for Disease Control and Prevention (CDC) has recommended that it is of vital importance for older hypertensive patients to receive an influenza vaccine during the 2020/21 influenza season [15], which is consistent with that announced by the US CDC [16] and the World Health Organization (WHO) [17]. Faced with the current events of coronavirus disease 2019 (COVID-19), the use of an influenza vaccine would help to reduce the overall burden on the healthcare system and save medical resources for the care of more COVID-19 patients [9,18]. In accordance with an Italian ecological study, the coverage rate of the influenza vaccination was associated with a reduced spread of COVID-19 [19]. Given that the outbreak of COVID-19 has resulted in severe morbidity and mortality in older adults, this becomes a much higher threat when combined with an influenza pandemic. Future work is required to raise the influenza vaccination coverage in older adults worldwide.

To the best of our knowledge, this large-size population-based study represents one of the first to estimate the recent influenza vaccination status among Chinese older adults with hypertension and to identify significant factors associated with actions toward vaccinations during the COVID-19 pandemic. These findings are expected to provide evidence on the underlying drivers of vaccine-related decision making among older individuals with hypertension. On this basis, more efficient strategies for influenza vaccinations could be developed for enhancing public health benefits.

## 2. Materials and Methods

This study formed part of an online large-scale cross-sectional investigation in October 2020 designed to evaluate vaccination coverage and the willingness toward influenza and pneumococcal vaccines together with their associated factors among older people in Shenzhen City of southern China. The research protocol was approved by the Longhua Center for Disease Control and Prevention, Shenzhen. Based on the literature review regarding the given topic, a panel consisting of three CDC researchers and two experts in epidemiology and infectious diseases was formed to develop the questionnaire for this investigation. The CDC researchers designed and prepared the first version of the questionnaire. Subsequently, a pre-test was conducted using fifty older adults other than those included in this study to assess the accuracy and consistency. After that, the first version of the questionnaire was reviewed again and modified by the CDC researchers and experts. The final version of the questionnaire consisted of four sections covering sociodemographic characteristics, health-related behaviors, vaccination coverage status, and the willingness to be vaccinated for influenza and pneumococcal pneumonia. All collected data were kept strictly confidential and only used for research purposes. The questionnaires were distributed through the most extensive online survey platform in China, Wen Juan Xing (Changsha Ranxing Information Technology Co., Ltd., Changsha, China). Detailed methods and part of the results of the cross-sectional survey have been published previously [20].

For this work, we collected data on elderly people over 60 years of age who reported having hypertension (*n* = 5216), similar to a relevant study [11]. Hypertension was defined as systolic blood pressure ≥ 140 mmHg and/or diastolic blood pressure ≥ 90 mmHg [21]. The disease information for each participant was not further confirmed via any medical records as it was an online anonymous survey. The study subjects gave responses to sociodemographic variables including gender (male, female), age (60–69, 70–79, or ≥80 years), marital status (married, divorced, or widowed), education level (middle school or below, high school, or college or above), living status (only with a partner, with children, or alone), monthly household income (<CNY 10,000, CNY 10,000–50,000, or >CNY 50,000), and house size (<50, 50–120, or >120 square meters). The variables of the health-related behaviors comprised use of a family doctor, frequency of physical examinations, smoking, drinking, and exercise. The information of monthly household income was presented in Chinese yuan (CNY) and US dollars (USD) with an average exchange rate of CNY 6.9 per USD in 2020, as already described [20]. The binary variables of the influenza vaccination were used as the dependent variable. Participants who were vaccinated during the latest influenza season were assigned to the vaccinated group and those who were not were classified into the unvaccinated group.

All of the variables were categorical and were summarized as frequencies and proportions. Pearson’s chi-squared tests or Fisher’s exact tests were performed as appropriate to estimate the statistical significance. Both bivariate and multivariable logistic regression analyses were undertaken to identify the independent predictors associated with an action toward the influenza vaccination by calculating the odds ratios (ORs) and 95% confidence intervals (CIs). Variables with a statistical significance in the bivariate analyses were brought into the multivariable analysis. All estimates were undertaken using IBM SPSS Statistics for Windows, version 25 (IBM Corp., Armonk, NY, USA). A two-sided *p*-value < 0.05 was considered statistically significant.

## 3. Results

Overall, 5216 older adults with hypertension were included this study. The descriptive characteristics are summarized in Table 1. The proportion of participants with an action of receiving an influenza vaccine was 4.7% (247/5216). Among the participants, 2736 (52.5%) were female, 3981 (76.3%) were aged between 60 and 69 years, and 3962 (76.0%) were married. Only 319 (6.1%) had an education level of college or above and the majority of cases (3844, 73.7%) reported living with their children. In parallel, around half of the participants (2801, 53.7%) had a monthly household income ranging from USD 1449 to USD 7246 and a majority (3903, 74.8%) lived in a house with a size from 50 to 120 square meters (m^2^). The influenza vaccination coverage was statistically significant between the subgroups according to age, education level, living status, monthly household income, and house size. In regard to health-related behaviors, greater than one quarter of participants (1465, 28.1%) reported the use of a family doctor and most (3558, 68.2%) indicated the frequency of health examination was less than once per year. Additionally, a majority of older adults never smoked (4269, 81.8%), never drank alcohol (4172, 80.0%), and exercised for more than half an hour each day (4119, 79.0%). The action on the influenza vaccination varied across the use of a family doctor, frequency of health examinations, and smoking. However, gender, marital status, drinking, and exercise were not statistically significant between the vaccinated and unvaccinated groups.

Using the action toward being vaccinated as the primary outcome, we employed in this study bivariate and multivariable logistic regression models to investigate the independent factors related to influenza vaccinations among older adults with hypertension. From the results of the bivariate regression analysis, the underlying factors that positively correlated with the influenza vaccination were age (OR 1.507, 95% CI: 1.117–2.033 for 70–79 years; OR 3.476, 95% CI: 2.183–5.534 for ≥ 80 years), education level (OR 1.529, 95% CI: 1.151–2.031 for high school; OR 2.241, 95% CI: 1.460–3.439 for college or above), living only with a partner (OR 1.484, 95% CI: 1.119–1.966), monthly household income >USD 7246 (OR 1.908, 95% CI: 1.308–2.783), house size >120 m^2^ (OR 3.007, 95% CI: 1.789–5.052), use of a family doctor (OR 2.681, 95% CI: 2.073–3.467), frequency of physical examinations (OR 2.771, 95% CI: 2.129–3.607 for 1–2 per year; OR 3.515, 95% CI: 1.879–6.575 for ≥ 3 per year), and non-smokers (OR 1.851, 95% CI: 1.234–2.779). However, gender, marital status, drinking, and exercise did not contribute to the influenza vaccination coverage and were rolled out from the multivariable analysis. The bivariate logistic analyses of the factors related to the influenza vaccination is presented in Table 2.

In the multivariable logistic regression analysis, the findings were generally similar to those of the bivariate models. Older adults with hypertension were more likely to receive an influenza vaccine aged 80 years and above (aOR 2.957, 95% CI: 1.784–4.900), with higher education levels (aOR 1.424, 95% CI: 1.060–1.914 for high school; aOR 1.681, 95% CI: 1.066–2.650 for college or above), and living only with a partner (aOR 1.432, 95% CI: 1.068–1.920). With respect to the health-related behaviors, the use of family doctors (aOR 2.275, 95% CI: 1.744–2.968) and a greater frequency of physical examinations (aOR 2.107, 95% CI: 1.601–2.772 for 1–2 per year; aOR 2.118, 95% CI: 1.083–4.143 for ≥ 3 per year) were related to an action toward an influenza vaccination. Smokers represented less possibility of a vaccination compared with non-smokers (aOR 1.829, 95% CI: 1.208–2.767). A forest plot depicting the multivariable logistic analysis of the independent predictors related to influenza vaccinations among older adults with hypertension is shown in Figure 1.

## 4. Discussion

The present survey lands at a critical period of the COVID-19 pandemic and aimed to investigate the influenza vaccination status in patients with hypertension who were older than 60 years. We identified the significant factors associated with an action toward an influenza vaccination. Generally, this study reported an extremely low influenza vaccination coverage rate (4.7%) in older people with hypertension, highlighting that there is a long way to go to fulfill the WHO and Chinese recommendations for influenza vaccination strategies in elderly people.

In contrast to previous studies in China, the coverage rate of influenza vaccinations obtained in this study was slightly higher than that reported in 2017 in Shanghai, China, where only 0.4% of patients with a chronic disease had received an influenza vaccine [10]. It was found to be close to the estimated annual influenza coverage (5.0–6.0%) among young workers in South China [22]. This result was also similar to the general population coverage in Guangzhou, China, ranging between 3.69% and 5.38% from 2011 to 2014 [22]. Compared with prior findings in other countries, this coverage was lower than that in Spain, where 65.6% of hospitalized people aged over 65 years had received an influenza vaccine; it was also much lower than that reported in American older people of more than 65 years of age during the 2018/19 influenza season [23]. According to the findings of a meta-analysis, recommendations by healthcare workers, a history of influenza vaccination, and the perceived safety and effectiveness of the vaccination are responsible for the lower values of influenza vaccination coverage in mainland China [24].

In the present study, older adults over 80 years of age were more likely to receive an influenza vaccination compared with the younger age groups, which was consistent with previously published results [25,26,27]. This might be because cases of hypertension are at a higher risk of influenza-related complications as the age increases, influencing their decision pertaining to immunization at the same time [28]. As expected, positive associations between the influenza vaccination and higher education levels were found, supporting related studies in various countries [29,30,31]. Participants obtaining a higher education degree might be associated with a greater understanding about influenza and its vaccine, ensuring correct preventive measures are taken to prevent infectious diseases [32,33,34]. Research has described that older participants refused to get vaccinated against influenza because they deemed that chronic respiratory disease was a contraindication for an influenza vaccination [7].

As already described in our work, living with others might lead to frequent social contact, giving these older individuals more possibilities to obtain vaccine or health-related information [7,20]. In this survey, participants living with a partner had more likelihood to get vaccinated than those living with children. This supports the demonstration of Ge et al., who indicated that older adults living with children might not be able to take care of themselves and thus be inconvenient to receive any vaccines in medical institutions [35]. In addition, several studies have also demonstrated the relationships between influenza vaccination coverage and economic factors [30]. The findings of this study found a higher vaccination coverage among older hypertensive patients with a higher monthly household income and living in a larger house, even if the relationships were not retained in the multivariable analysis.

Healthcare workers play a key role in promoting the influenza vaccination among various populations [36,37,38,39,40]. Previous studies have identified that older people expressed more willingness to get vaccinated under the recommendations of doctors [41,42]. In this study, it was found that the use of a family doctor and a greater frequency of physical examinations were significantly correlated with an action toward the influenza vaccination. This supports other studies that linked a higher probability of an influenza vaccination with a greater use of health services [31,43]. Subjects who undergo physical examinations and use medical services in any way present higher vaccination coverage rates, given by a greater interest in the early prevention of diseases and health-promoting behaviors [12]. A lack of initiative around suggestions from a clinician or family doctor might be an important factor in the low vaccination rate in China [36]. The present findings indicated that smoking, which belongs to a bad life habit, was another independent predictor within the health-related behaviors. In line with prior studies, higher vaccination rates were found in non-smokers than in former or current smokers [12,44]. It is likely that people with healthy lifestyles tend to seek healthy and preventive behaviors and therefore have a higher rate of interest in the influenza vaccination [12,44].

In China, all populations aged over 6 months are recommended to get vaccinated before the influenza season because of the national conditions of a large population density, a high turnover, and the currently unsatisfied vaccination coverage status [22]. In light of the guidelines and consensus pertaining to recommendations for influenza vaccinations in elderly people in China, adults aged over 60 years (particularly with cardiovascular illnesses) should receive an influenza vaccination annually because influenza viruses are prone to rapid antigenic mutation [1]. Evidence has pointed out that coinfection with the influenza virus may worsen the clinical consequence of COVID-19 patients, which will increase the hospital stress for the National Health System [18]. Finally, individuals with hypertension are not allowed to accept a COVID-19 vaccine based on the current strategies against COVID-19 in China. As such, improvements on influenza vaccine coverage will contribute to the health of older people with hypertension as well as reduce the disease burden and alleviate the stress of public health institutions during the outbreak of COVID-19.

A few limitations should be taken into account. First, the influenza vaccination status and hypertension conditions were investigated using a self-reported questionnaire. Second, the study population only consisted of older adults from a single center that was not representative of all elderly people in China.

## 5. Conclusions

The main independent predictors of the action of participants toward influenza vaccination were a younger age, a lower education level, a lack of a family doctor, less frequency of physical examinations, and smoking. These findings ought to be used for health authorities to formulate suitable interventions to enhance influenza control and prevention.

## Figures and Tables

**Figure 1 vaccines-09-01105-f001:**
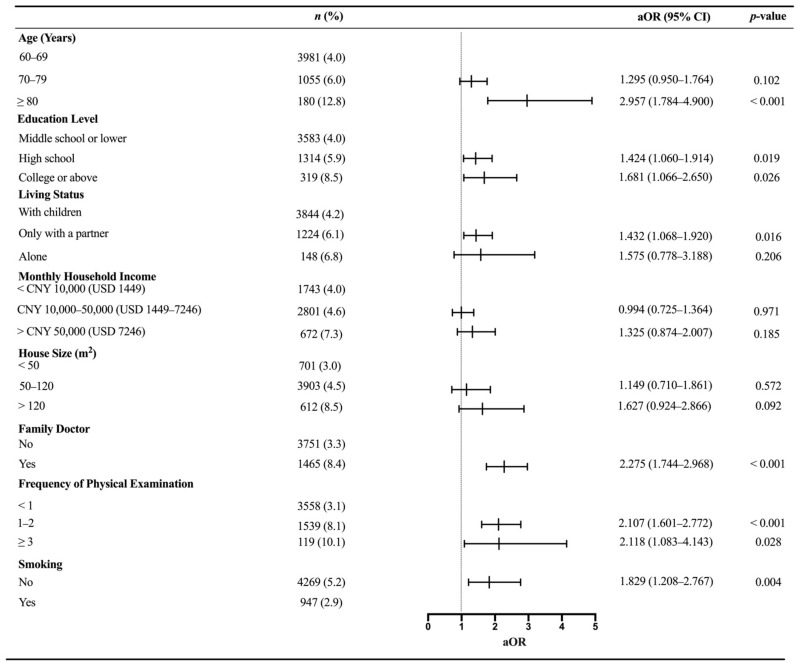
Factors for influenza vaccinations among the elderly with hypertension. aOR: adjusted odd ratio; CI: confidence interval. *n* indicates the numbers of each category. % indicates the influenza coverage rates. Variables included in the multivariable logistic regression analyses model after adjustments for age, education level, living status, monthly household income, house size, use of a family doctor, frequency of physical examinations, and smoking.

**Table 1 vaccines-09-01105-t001:** Characteristics of the study population.

			Influenza Vaccination		
	*n* (%)	95% CI	Vaccinated (%)	Unvaccinated (%)	*p*-Value
Total	5216		247 (4.7)	4969 (95.3)	
*Sociodemographics*					
Gender					0.332
Male	2480 (47.5)	46.2–48.9	110 (44.5)	2370 (47.7)	
Female	2736 (52.5)	51.1–53.8	137 (55.5)	2599 (52.3)	
Age (years)					<0.001
60–69	3981 (76.3)	75.2–77.5	161 (65.2)	3820 (76.9)	
70–79	1055 (20.2)	19.1–21.3	63 (25.5)	992 (20.0)	
≥80	180 (3.5)	3.0–3.9	23 (9.3)	157 (3.2)	
Marital Status					0.288
Married	3962 (76.0)	74.8–77.1	184 (74.5)	3778 (76.0)	
Divorced	84 (1.6)	1.3–2.0	7 (2.8)	77 (1.5)	
Widowed	1170 (22.4)	21.3–23.6	56 (22.7)	1114 (22.4)	
Education Level					<0.001
Middle school or lower	3583 (68.7)	67.4–70.0	142 (57.5)	3441 (69.2)	
High school	1314 (25.2)	24.0–26.4	78 (31.6)	1236 (24.9)	
College or above	319 (6.1)	5.5–6.8	27 (10.9)	292 (5.9)	
Living Status					0.012
With children	3844 (73.7)	72.5–74.9	162 (65.6)	3682 (74.1)	
Only with a partner	1224 (23.5)	22.3–24.6	75 (30.4)	1149 (23.1)	
Alone	148 (2.8)	2.4–3.3	10 (4.0)	138 (2.8)	
Monthly Household Income					0.002
<CNY 10,000 (<USD 1449)	1743 (33.4)	32.1–34.7	69 (27.9)	1674 (33.7)	
CNY 10,000–50,000 (USD 1449–7246)	2801 (53.7)	52.3–55.1	129 (52.2)	2672 (53.8)	
>CNY 50,000 (>USD 7246)	672 (12.9)	12.0–13.8	49 (19.8)	623 (12.5)	
House Size (m^2^)					<0.001
<50	701 (13.4)	12.5–14.4	21 (8.5)	680 (13.7)	
50–120	3903 (74.8)	73.6–76.0	174 (70.4)	3729 (75.0)	
>120	612 (11.7)	10.9–12.6	52 (21.1)	560 (11.3)	
*Health-Related Behaviors*					
Family Doctor					<0.001
No	3751 (71.9)	70.7–73.1	124 (50.2)	3627 (73.0)	
Yes	1465 (28.1)	26.9–29.3	123 (49.8)	1342 (27.0)	
Frequency of Physical Examinations					<0.001
<1	3558 (68.2)	66.9–69.5	110 (44.5)	3448 (69.4)	
1–2	1539 (29.5)	28.3–30.7	125 (50.6)	1414 (28.5)	
≥3	119 (2.3)	1.9–2.7	12 (4.9)	107 (2.2)	
Smoke					0.003
No	4269 (81.8)	80.8–82.9	220 (89.1)	4049 (81.5)	
Yes	947 (18.2)	17.1–19.2	27 (10.9)	920 (18.5)	
Drink					0.226
No	4172 (80.0)	78.9–81.1	205 (83.0)	3967 (79.8)	
Yes	1044 (20.0)	18.9–21.1	42 (17.0)	1002 (20.2)	
Exercise					0.152
No	1097 (21.0)	19.9–22.1	43 (17.4)	1054 (21.2)	
Yes	4119 (79.0)	77.9–80.1	204 (82.6)	3915 (78.8)	

CI: confidence interval.

**Table 2 vaccines-09-01105-t002:** Factors for influenza vaccination among the elderly with hypertension.

		Bivariate		
	*n* (%)	OR	95% CI	*p*-Value
*Sociodemographics*				
Gender				
Male	2480 (4.4)		Ref.	
Female	2736 (5.0)	1.136	0.878–1.469	0.332
Age (Years)				
60–69	3981 (4.0)		Ref.	
70–79	1055 (6.0)	1.507	1.117–2.033	0.007
≥80	180 (12.8)	3.476	2.183–5.534	<0.001
Marital Status				
Married	3962 (4.6)		Ref.	
Divorced	84 (8.3)	1.867	0.849–4.104	0.120
Widowed	1170 (4.8)	1.032	0.760–1.402	0.840
Education Level				
Middle school or lower	3583 (4.0)		Ref.	
High school	1314 (5.9)	1.529	1.151–2.031	0.003
College or above	319 (8.5)	2.241	1.460–3.439	<0.001
Living Status				
With children	3844 (4.2)		Ref.	
Only with a partner	1224 (6.1)	1.484	1.119–1.966	0.006
Alone	148 (6.8)	1.647	0.851–3.189	0.139
Monthly Household Income				
<CNY 10,000 (<USD 1449)	1743 (4.0)		Ref.	
CNY 10,000–50,000 (USD 1449–7246)	2801 (4.6)	1.171	0.869–1.579	0.299
>CNY 50,000 (>USD 7246)	672 (7.3)	1.908	1.308–2.783	0.001
House Size (m^2^)				
<50	701 (3.0)		Ref.	
50–120	3903 (4.5)	1.511	0.954–2.394	0.079
>120	612 (8.5)	3.007	1.789–5.052	<0.001
*Health-Related Behaviors*				
Family Doctor				
No	3751 (3.3)		Ref.	
Yes	1465 (8.4)	2.681	2.073–3.467	<0.001
Frequency of Physical Examination				
<1	3558 (3.1)		Ref.	
1–2	1539 (8.1)	2.771	2.129–3.607	<0.001
≥3	119 (10.1)	3.515	1.879–6.575	<0.001
Smoking				
No	4269 (5.2)	1.851	1.234–2.779	0.003
Yes	947 (2.9)		Ref.	
Drinking				
No	4172 (4.9)	1.233	0.878–1.731	0.226
Yes	1044 (4.0)		Ref.	
Exercise				
No	1097 (3.9)		Ref.	
Yes	4119 (5.0)	1.277	0.913–1.787	0.153

OR: odds ratio; CI: confidence interval; Ref., reference. *n* indicates the numbers of each category. % indicates the influenza coverage rates.

## Data Availability

Access to the data presented in this study can be acquired by connecting to the corresponding authors via email. The data are not publicly available due to restrictions of privacy.

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
