# Peer review of "Influenza Vaccination Coverage among Older Adults with Hypertension in Shenzhen, China: A Cross-Sectional Survey during the COVID-19 Pandemic"

_vaccines, 2021, doi:10.3390/vaccines9101105_

Round 1
Reviewer 1 Report
The paper deals with Influenza Vaccination Coverage among Older Adults with Hypertension in Shenzhen, China: A Cross-Sectional Survey during the COVID-19 Pandemic. Please see my suggestions for improving the shape and the content of the manuscript:
2. Material and Method
Please detail: who made the surveys? who validated them? there were some collaboration with sociologists, specialists in such questionnaires? were these questionnaires pre-tested before their application to all respondents? based on which criteria the items were chosen/ how do you have chosen/decided the optimal items? based on which criteria, the respondents were chosen? etc.
L180-182. This phrase must be moved at the final of Introduction section, as it belongs to the aim of the study and underline very good the novelty of the research.
Please avoid using the personal manner of addressing “we”, “our”, (it is annoying so many “we”) and use the impersonal one; the text will sound much more professional. Please justify the text format in the manuscript, in all sections.Author Response
Dear Reviewer,
Thank you very much for your concern on our manuscript titled “Influenza Vaccination Coverage among Older Adults with Hypertension in Shenzhen, China: A Cross-Sectional Survey during the COVID-19 Pandemic”. Your professional suggestions and comments are of vital benefit to enhancing the manuscript’s quality and our future work. We have made some changes according to the comments and tried our best to improve the manuscript. Please see our point-by-point responses as listed below.
We appreciate your warm work earnestly and hope that the revisions will meet with approval.
Yours sincerely,
Qing Chen
Point 1: Please detail: who made the surveys? who validated them? there were some collaboration with sociologists, specialists in such questionnaires? were these questionnaires pre-tested before their application to all respondents? based on which criteria the items were chosen/ how do you have chosen/decided the optimal items? based on which criteria, the respondents were chosen? etc.
Response 1: Thank you very much for your comments. We are sorry for the unclear description of the methodology. Based on the literature review regarding the given topic, a panel consisting of three Shenzhen CDC researchers and two experts in Southern Medical University was formed to develop the questionnaire for this investigation. CDC researchers designed and prepared the first version of the questionnaire. Subsequently, we conducted a pre-test using fifty older adults other than those included in this study to assess the accuracy and consistency. After that, the first version of the questionnaire was reviewed again and modified by CDC researchers and experts. The final version of the questionnaire consisted of four sections covering sociodemographic characteristics, health-related behaviors, vaccination coverage status, and willingness to be vaccinated for influenza and pneumococcal pneumonia. Regarding the item selection criteria, we calculated the Cronbach’s alpha coefficients for each section, and we chose the items with a value ≥ 0.6 in the final questionnaire. Regarding the respondent selection criteria, only those older adults who consent to be a part of the present study would be recruited. Also, respondents who took less than 100 seconds to complete the questionnaire would be excluded. All collected data would be kept strictly confidential and only be used for research purposes. We have included the statement in the revised manuscript. (Page 2-3, Line 103-122)
Point 2: L180-182. This phrase must be moved at the final of the Introduction section, as it belongs to the aim of the study and underline very good the novelty of the research.
Response 2: Many thanks for your suggestion. We have moved the phrase at the end of the Introduction section in the revised manuscript. (Page 2, Line 90-93)
Point 3: Please avoid using the personal manner of addressing “we”, “our”, (it is annoying so many “we”) and use the impersonal one; the text will sound much more professional. Please justify the text format in the manuscript, in all sections.
Response 3: We are sorry that we use so many personal manners of “we” and “our”. Thank you very much again for the comments. We have changed them throughout the revised manuscript and have justified the text format in all sections of the revised manuscript.
Reviewer 2 Report
- In Table 2, the authors should use as the reference category in each variable the group with the lowest percentage of vaccinated. As an example, for the gender variable, the reference group is women. It should be males since they are the ones with the lowest proportion of vaccinated patients. Thus the OR would be 1.136 (95% CI 0.878-1.469) with males as the reference category.
- In Table 2, in the values of each variable, N is given and percentage, but the % refers to the proportion of each category in the sample.
Thus instead of
Male 2480 (47.5)
Female 2736 (52.5)
The % should be the proportion of vaccinated in each category
Male 2480 (4.4)
Female 2736 (5.5)
These changes should be done throughout the table
Author Response
Dear Reviewer,
Thank you very much for your concern on our manuscript titled “Influenza Vaccination Coverage among Older Adults with Hypertension in Shenzhen, China: A Cross-Sectional Survey during the COVID-19 Pandemic”. Your professional suggestions and comments are of vital benefit to enhancing the manuscript’s quality and our future work. We have made some changes according to the comments and tried our best to improve the manuscript. Please see our point-by-point responses as listed below.
We appreciate your warm work earnestly and hope that the revisions will meet with approval.
Yours sincerely,
Qing Chen
Point 1: In Table 2, the authors should use as the reference category in each variable the group with the lowest percentage of vaccinated. As an example, for the gender variable, the reference group is women. It should be males since they are the ones with the lowest proportion of vaccinated patients. Thus the OR would be 1.136 (95% CI 0.878-1.469) with males as the reference category.
Response 1: Thank you very much for your comment and this clear example. According to this comment, we have changed the reference category in each variable in the revised manuscript.
Point 2: In Table 2, in the values of each variable, N is given and percentage, but the % refers to the proportion of each category in the sample. Thus instead of “Male 2480 (47.5)” and “Female 2736 (52.5)”, the % should be the proportion of vaccinated in each category: “Male 2480 (4.4)” and “Female 2736 (5.5)”. These changes should be done throughout the table.
Response 2: Many thanks for your comment again. We have used the proportion of vaccinated in each category in the revised Table 2. Also, the corresponding values in the text were changed in the revised manuscript.
Round 2
Reviewer 1 Report
The authors responded to all my requests.